# LOW-RANK ATTENTION AND CONTRASTIVE ALIGNMENT FOR DEEP MULTI-VIEW CLUSTERING

## ABSTRACT

Recent years have witnessed significant advancements in deep multi-view clustering (MVC). However, prevailing methods exhibit three critical limitations: (1) poor scalability for large-scale datasets, (2) neglect of anchor semantic consistency in feature alignment, and (3) inability to capture high-order feature interactions. To overcome these challenges, we propose a Low-Rank Attention and Contrastive Alignment framework (LRACA). Unlike conventional approaches that align sample-level features in shared subspaces, LRACA employs a category-aware anchor generation module to directly align high-level semantic prototypes (i.e., category centers) across views, explicitly enforcing clustering semantic consistency. Furthermore, we devise a dynamic low-rank attention mechanism to enhance feature discriminability, where entropy regularization constrains attention weight distributions to derive clustering pseudo-labels. Finally, a pseudo-label-guided cluster-level contrastive learning module maximizes cross-view mutual information through a feed-forward optimization paradigm. Extensive experiments on six large-scale multi-view datasets demonstrate that LRACA significantly outperforms state-of-the-art methods.

## 1 INTRODUCTION

In this era of explosive information growth, it has become increasingly imperative to effectively harness and synthesize multifaceted data streams, extracting and reconstructing them into actionable insights that can inform and guide more rational human decision-making and behavioral patterns Qin et al. (2022b); Peng et al. (2022); Jia et al. (2023); Qin et al. (2022a); Huang et al. (2024). The integration and exploration of multi-view information has garnered more and more attention in recent years Sun (2013); Hu et al. (2019); Wei et al. (2020). In multi-view learning, clustering algorithms have emerged as a research focal point due to their performance in unsupervised learning scenarios with unlabeled samples. However, Current research exhibits notable limitations in balancing cross-view heterogeneity alignment with computational efficiency. Conventional shallow models (e.g., subspace learning Shang et al. (2023), multi-kernel fusion Long et al. (2024)) establish view correlations through linear assumptions or kernel function due to limited representational capabilities, are difficult to model complex nonlinear relationships. Wang et al. (2021). While deep learning approaches enhance feature expressiveness via nonlinear mappings—exemplified by contrastive learning-based MFLVC Xu et al. (2022) that strengthens consistency through cross-view instance alignment—their rigid similarity constraints suppress view-specific discriminative information. Scalable anchor-sampling MVC methods Xia et al. (2022) reduce computation via random core sample selection, yet distributionally deviant anchors introduce semantic bias, undermining cross-view alignment. Furthermore, low-rank approximation attention mechanisms (e.g. Linformer Wang et al. (2020)) achieve linear complexity, but this may lead to a decline in the ability to distinguish characteristics through dimensionality reduction strategies decoupled from clustering objectives. These issues collectively highlight the unresolved balance between semantic alignment precision and computational efficiency in heterogeneous, large-scale MVC scenarios.

To address these challenges, we propose we propose a Low-Rank Attention and Contrastive Alignment framework (LRACA). The methodology's core innovation lies in resolving view heterogeneity alignment and computational bottlenecks through category-aware anchor sampling and low-rank adaptive attention modules. Specifically, during pre-training, we employ label-driven K-means algorithms to select category-representative anchors, constructing cross-view consistent anchor graphs

to avoid semantic drift from random sampling. Subsequently, we design dynamic linear-complexity self-attention modules enhanced by entropy regularization to constrain attention weight distributions, simultaneously reducing computational overhead and amplifying feature discriminability. Ultimately, a hierarchical contrastive learning framework achieves synergistic enhancement of efficiency and precision through phase-wise optimization of intra-view private information and cross-view semantic consistency under anchor graph guidance. Our methodology features three key innovations:

- We propose a novel category-aware anchor sampling strategy integrated with a learnable low-rank projection matrix. Anchor-guided sparsification enhance feature quality and the projection dimension can be further reduced, providing high-confidence guidance and overcoming the information loss of traditional approximation methods.

- We develop an efficient contrastive learning framework that replaces instance-wise full-sample comparisons with cluster-level alignment. Through attention-guided semantic matching, this approach not only enhances inter-view consistency and complementary information integration, but also reduces computational redundancy while maintaining feature discriminability.

- Experimental results on six common large-scale multi-view datasets demonstrate that LRACA significantly improves clustering performance under mainstream clustering valuation metrics compared to current state of-the-art methods.

## 2 RELATED WORK

### 2.1 LARGE-SCALE MULTI-VIEW CLUSTERING

Multi-view clustering handles data from diverse perspectives. Conventional methods include graph-based Tao et al. (2023), subspace Shang et al. (2023), NMF Zhao et al. (2020), and kernel-based Long et al. (2024) techniques, with deep learning (e.g., autoencoders Zhang et al. (2019), GNNs Xia et al. (2022); Wang et al. (2019)) gaining traction. To manage large-scale data, anchor-based MVC methods reduce complexity by selecting anchor points per view to build graphs Li et al. (2019; 2015). Methods like CGMSC Liu et al. (2021b) unify graph fusion and anchor learning. However, misaligned anchors across views degrade fusion and clustering. While AUP Wang et al. (2022) addresses alignment, its iterative optimization is costly. Recent works focus on graph refinement (SURER Wang et al. (2024), BF-CGF Yang et al. (2024)) and efficiency (parameter-free fusion Duan et al. (2024)). Inspired by these, our LRACA uniquely integrates dynamic low-rank projection with category-aware anchor sampling to enhance cross-view consistency efficiently.

### 2.2 CONTRASTIVE LEARNING

Contrastive learning, maximizing similarity for positive pairs and minimizing for negatives Oord et al. (2018); Tian et al. (2020), has driven progress in multi-view representation learning Lin et al. (2022); Liu et al. (2023); Yang et al. (2022). However, prevalent approaches often focus on single views with artificial augmentations, incurring overhead and potential semantic inconsistency. They also struggle to capture high-order semantic correlations organically integrating reconstruction and consistency. This work elevates contrastive learning to the cluster level, mitigating inter-view semantic discrepancies via pseudo-label-guided cross-view mutual information maximization.

## 3 METHODOLOGY

In this paper, we propose an end-to-end deep multi-view clustering framework to provide robust network structure and align semantic consistency across views for improving the performance of clustering. Fig. 1 illustrates the framework of LRACA.

### 3.1 ANCHOR-GUIDED LOW-RANK ATTENTION MODULE

We combine anchor points with a dynamic low-rank attention mechanism to form an anchor-guided linear self-attention module. The core function of anchor points is to improve feature quality, thereby

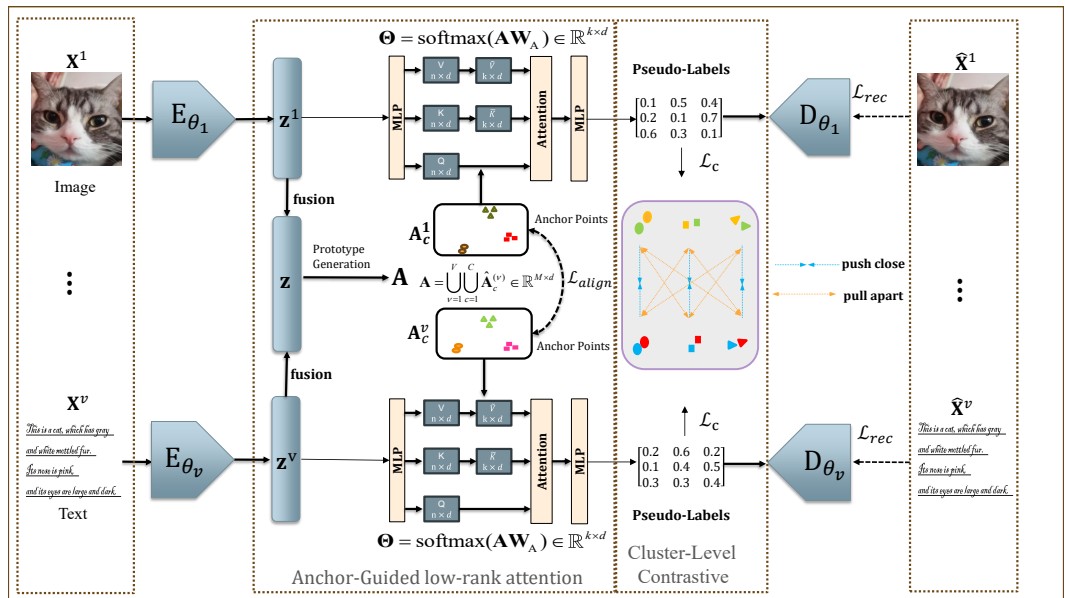

Figure 1: The framework of LRACA. It integrates an anchor-guided linear self-attention module and a cluster-level contrastive learning module. Specifically, the view-specific encoder synergizes feature clustering to dynamically generate semantic-aware anchor points and initialize pseudo-labels for each view. The enhanced feature representation is constructed by concatenating the original input $X^v$ with the generated anchor set $A^v$, which is subsequently fed into the linear self-attention mechanism to refine discriminative embeddings and iteratively update pseudo-labels. Pseudo-labels drive the cluster-level contrastive learning module to align semantic consistency across views, ultimately yielding robust multi-view clustering results.

allowing for more aggressive low-rank approximation (reducing rank r) and indirectly reducing computational complexity. The attention mechanism dynamically allocates weights by calculating the similarity between samples and anchor points, avoiding fine-grained calculations on the entire sample. The Anchor-guided View Encoder addresses two pivotal challenges in multi-view clustering: cross-view semantic discrepancies and the prohibitive computational complexity of full-sample alignment. Conventional methodologies frequently struggle to preserve view-specific discriminative features while aligning heterogeneous distributions. Our key innovation lies in the label-driven alignment mechanism, which enhances semantic coherence through meticulously designed alignment loss functions and pseudo-label generation, ensuring congruence among latent cluster centers across different views for identical categories.

For the $v$-th view input $\mathbf{X}^v \in \mathbb{R}^{d_v \times N}$, a view-specific encoder $E_{\theta_v}$ maps it to latent features $\mathbf{Z}^v = E_{\theta_v}^{(}\mathbf{X}^v) \in \mathbb{R}^{N \times d}$, where $d$ is the shared latent dimension. A decoder $D_{\phi_v}$ reconstructs the input as $\hat{\mathbf{X}}^v = D_{\phi_v}(\mathbf{Z}^v)$, optimized via the reconstruction loss:

$$\mathcal{L}_{\text{recon}} = \sum_{v=1}^{V} \|\mathbf{X}^v - D_{\phi_v}(E_{\theta_v}(\mathbf{X}^v))\|_F^2 \tag{1}$$

The fused latent representation combines multi-view features through concatenation:

$$\mathbf{Z}_{\text{fusion}} = \left[\mathbf{Z}^{(1)}\|\mathbf{Z}^{(2)}\|\cdots\|\mathbf{Z}^{(v)}\right] \in \mathbb{R}^{N \times (v \cdot d)} \tag{2}$$

which preserves view-specific discriminative patterns while enabling cross-view interaction. The fused features are then clustered to generate pseudo-labels $\mathbf{Y}$, which iteratively guide anchor refinement.

The fused latent representation $\mathbf{Z}_{\text{fusion}}$ serves as the foundation for generating pseudo-labels that guide cross-view semantic alignment. A two-stage process ensures robust cluster formation: K-

means clustering is applied to $\mathbf{Z}_{\text{fusion}}$ to obtain initial pseudo-labels:

$$\mathbf{Y}_{\text{pseudo}} = \text{K-means}(\mathbf{Z}_{\text{fusion}}, C) \in \{1, \ldots, C\}^N \tag{3}$$

where $C$ denotes the number of clusters. For each cluster $c \in \{1, \ldots, C\}$ and view $v$, samples belonging to cluster $c$ are identified in the shared latent space:

$$\mathcal{S}_k^{(c)} = \left\{ \mathbf{z}_i^{(v)} \mathbf{y}_{\text{pesudo},i} = c \right\} \tag{4}$$

where $y_{\text{pesudo},i}$ is the refined pseudo-label. Sub-clustering via K-means on $\mathcal{S}_c^{(v)}$ yields $m$ fine-grained anchors:

$$\mathbf{A}_c^{(v)} = \text{K-means}(\mathbf{S}_c^{(v)}, m) \in \mathbb{R}^{m \times d} \tag{5}$$

To enforce semantic agreement across views, cluster centroids $\mathbf{A}_c^{(v)}$ are decoded to view-specific anchors $\hat{\mathbf{A}}_c^{(v)} = g_{\phi_v}(\mathbf{A}_c^{(v)})$ and aligned via:

$$\mathcal{L}_{\text{align-anchor}} = \sum_{v \neq u} \sum_{k=1}^{K} \left\| \mathbf{A}_c^{(v)} - \mathbf{A}_c^{(u)} \right\|_F^2 \tag{6}$$

where $\| \cdot \|_F^2$ ensures distributional consistency in the latent space. This alignment guarantees that anchors representing the same semantic category are invariant to view-specific perturbations.

The total training objective integrates reconstruction fidelity, latent distribution alignment, and anchor consistency:

$$\mathcal{L}_{pre} = \mathcal{L}_{\text{recon}} + \mathcal{L}_{\text{align-anchor}} \tag{7}$$

This unified framework creates a self-reinforcing cycle: pseudo-labels guide anchor generation, while aligned anchors refine pseudo-labels through contrastive learning.

Pseudo-labels $\mathbf{Y}_{\text{pseudo}}$ are iteratively updated using both feature similarity and anchor consistency, reducing label noise. Sub-clustering within each pseudo-cluster ($\mathbf{S}_c^{(v)}$) captures intra-class diversity while maintaining cross-view alignment.

Traditional self-attention mechanisms suffer from quadratic complexity $\mathcal{O}(N^2)$, which costs large Computility for multi-view data. We propose a *dynamic low-rank decomposed attention* mechanism that achieves linear complexity $\mathcal{O}(Nk)$ ($k \ll N$) while enhancing cross-view semantic alignment. The proposed method introduces three fundamental improvements over baseline approaches: (1) Dynamic Projection, which employs a learnable projection matrix $\mathbf{\Theta}$ to dynamically adapt to data distributions, in contrast to rigid Johnson-Lindenstrauss (JL) projections; (2) Semantic Anchoring, which explicitly couples cross-view anchors to preserve semantic consistency across views, replacing error-prone random anchor initialization; and (3) Entropy Regularization, where the entropy constraint prevents degenerate solutions where attention focuses solely on a few dominant anchors. Given input features $\mathbf{X}^v \in \mathbb{R}^{N \times d}$ and anchor prototypes

$$\mathbf{A} = \bigcup_{\nu=1}^{V} \bigcup_{c=1}^{C} \hat{\mathbf{A}}_c^{(\nu)} \in \mathbb{R}^{M \times d} \tag{8}$$

, where $M = V * C * m$, we construct dynamic projection parameters through anchor semantic fusion:

$$\mathbf{\Theta} = \text{softmax}(\mathbf{A}\mathbf{W}_c) \in \mathbb{R}^{k \times d} \tag{9}$$

where $\mathbf{W}_c \in \mathbb{R}^{d \times k}$ is learnable. Specifically, the dynamic low rank projection matrix $\mathbf{\Theta}$ uses fused clustering centers $\mathbf{A}$ to dynamically project features into the k-dimensional semantic subspace, replacing JL projection and initializing through C's SVD principal component. Meanwhile, softmax normalizes by row to ensure that each row (i.e., each low dimension basic vector) is a weighted combination of anchor semantics. For each attention head:

$$\mathbf{Q} = \mathbf{X}^v \mathbf{W}_Q \in \mathbb{R}^{N \times d},$$

$$\widetilde{\mathbf{K}} = \mathrm{softmax}(\mathbf{X}^v \mathbf{\Theta}^\top)^\top \mathbf{X}^v \in \mathbb{R}^{k \times d}, \tag{10}$$

$$\widetilde{\mathbf{V}} = \mathrm{softmax}(\mathbf{X}^v \mathbf{\Theta}^\top)^\top \mathbf{X}^v \in \mathbb{R}^{k \times d}. \tag{11}$$

The attention weights are computed as:

$$\mathbf{Attention}(\mathbf{Q}, \widetilde{\mathbf{K}}, \widetilde{\mathbf{V}}) = \mathrm{softmax}\left(\frac{\mathbf{Q}\widetilde{\mathbf{K}}^\top}{\sqrt{d}}\right)\widetilde{\mathbf{V}} \in \mathbb{R}^{N \times k} \tag{11}$$

Among them $\mathbf{W}_Q$ is the query matrix after linear transformation. The query only comes from the original input features; Keys and values come from the enhanced feature matrix, introducing anchor information. Each attention head has an independent weight matrix for learning multiple relationships, where $d$ is the dimension of the key and serves as a hyperparameter. By matrix multiplication, the original high-dimensional feature $X$ is projected onto a low dimensional space to probabilistically represent the semantic association strength between the i-th sample and the j-th anchor point. Then, the original feature $X^v$ is weighted and summed using weight matrix to obtain a compressed low rank matrix, which serves as the basis for subsequent attention calculations. The attention weight summarizes the semantics of $N$ samples using $k$, thereby reducing the cost.

### 3.1.1 Entropy Regularization

The entropy constraint prevents degenerate solutions where attention focuses solely on a few dominant anchors.

$$\mathcal{L}_{\mathrm{ent}} = -\frac{1}{N}\sum_{i=1}^{N}\sum_{j=1}^{N+M} a_{ij} \log a_{ij} \tag{12}$$

where $a_{ij}$ denotes the attention weight between query $i$ and key $j$. This encourages sparsity for discriminative features while maintaining diversity.

The final output combines multi-head results with parametric skip-connection:

$$\mathbf{Z_o} = \mathrm{LayerNorm}(\mathbf{X} + \mathrm{Concat}(\mathbf{head}_1, ..., \mathbf{head}_h)\mathbf{W}_o) \tag{13}$$

where the $\mathbf{X} \in \mathbb{R}^{N \times d}$ is the input sequence (sequence length $N$, feature dimension $d$), the $\mathbf{head}_i \in \mathbb{R}^{N \times d/h}$ is the output of the $i$-th attention head ($h$ = number of heads)and the $\mathbf{W}_o \in \mathbb{R}^{d \times d}$ indicates the output projection matrix for multi-head concatenation.

### 3.2 Cluster-Level Contrastive Learning with Pseudo-Label Guidance

Traditional contrastive learning in multi-view clustering often suffers from sample-level noise due to inconsistent pseudo-labels across views. To address this, we propose a cluster-level contrastive learning paradigm that operates on cluster probability vectors instead of raw features. Cluster centroids provide stabilized representations of semantic categories. The key innovation lies in leveraging cluster with pseudo-labels to define view-invariant positive/negative pairs, thereby aligning semantic structures across heterogeneous views while suppressing label noise. For each sample $i$ across $V$ views, let $\mathbf{p}_i^v \in \mathbb{R}^K$ denote its cluster probability vector in the $v$-th view, derived from pseudo-labels $Y_{\mathrm{pseudo}}$. Positive pairs are defined as probability vectors of the *same sample* across different views $(\mathbf{p}_i^{v_1}, \mathbf{p}_i^{v_2})$, while negative pairs include *different samples* regardless of views $(\mathbf{p}_i^{v_1}, \mathbf{p}_j^{v_2}), j \neq i$. This strategy enhances intra-cluster compactness and inter-cluster separability. The similarity between two probability vectors is measured via cosine similarity Chen et al. (2020):

$$s(\mathbf{p}_i^{v_1}, \mathbf{p}_j^{v_2}) = \frac{(\mathbf{p}_i^{v_1})^\top \mathbf{p}_j^{v_2}}{\|\mathbf{p}_i^{v_1}\|\|\mathbf{p}_j^{v_2}\|} \tag{14}$$

---

**Algorithm 1** LRACA Algorithm

---

**Require:** Multi-view dataset $\mathcal{X} = \{\mathbf{X}_v\}_{v=1}^V$, cluster number $K$, rank $r$, temperature $\tau$, hyperparameters $\lambda_c$, $\lambda_{\text{ent}}$
**Ensure:** Clustering labels $\mathbf{Y}$
 1: **Initialization:**
 2: Initialize encoder weights $\theta_v$, decoder weights $\phi_v$ via Xavier initialization
 3: Initialize anchor set $\mathbf{A}$ with K-means on concatenated features
 4: Initialize low-rank projection $\boldsymbol{\Theta}$ via anchor cluster centers
 5: **Pretraining Phase:**
 6: **for** epoch = 1 to $E_{\text{pre}}$ **do**
 7:     update anchor alignment loss $\mathcal{L}_{\text{align-anchor}}$ via Eq.equation 6
 8:     Calculate entropy regularization $\mathcal{L}_{\text{ent}}$ via Eq.equation 12
 9: **end for**
10: **Contrastive Fine-tuning:**
11: **for** epoch = 1 to $E_c$ **do**
12:     Calculate contrastive loss $\mathcal{L}_c$ via Eq.equation 15
13: **end for**
14: **return** $\mathbf{Y}$

---

The cluster-level contrastive loss for view pair $(v_1, v_2)$ is formulated as:

$$\mathcal{L}_c(\mathbf{v}_1, \mathbf{v}_2) = -\frac{1}{N}\sum_{i=1}^N \log \frac{\exp\left(s(\mathbf{p}_i^{v_1}, \mathbf{p}_i^{v_2})/\tau\right)}{A_i + B_i},$$

$$\text{where} \quad A_i = \sum_{j=1}^N \exp\left(s(\mathbf{p}_i^{v_1}, \mathbf{p}_i^{v_2})/\tau\right), \tag{15}$$

$$B_i = \sum_{\substack{j=1 \\ j \neq i}}^N \exp\left(s(\mathbf{p}_j^{v_1}, \mathbf{p}_j^{v_2})/\tau\right).$$

where $\tau > 0$ is a temperature hyperparameter. The denominator contrasts positive pairs against all negatives, explicitly maximizing mutual information for consistent clusters.

The total loss integrates reconstruction, attention regularization, and contrastive objectives:

$$\mathcal{L} = \mathcal{L}_{\text{rec}} + \lambda_{ent}\mathcal{L}_{\text{ent}} + \lambda_c\mathcal{L}_c \tag{16}$$

where $\lambda_{ent}, \lambda_c$ balance cross-view alignment and cluster discrimination.

### 3.3 COMPLEXITY ANALYSIS

Let $m$, $n_v$, $d_v$, $K$, $r$, $T_{\text{kmeans}}$, and $h$ denote the batch size, number of views, feature dimension per view, cluster count, low-rank projection dimension, K-means iterations, and attention heads respectively.

The dynamic low-rank attention mechanism achieves linear complexity through three key phases:via (Eq. 9) construct dynamic projection matrix $\boldsymbol{\Theta} \in \mathbb{R}^{k \times d}$ with $O(dk)$ complexity, via (Eq. 10) compute $\widetilde{\mathbf{K}} = \text{softmax}(\mathbf{X}\boldsymbol{\Theta}^\top)^\top \mathbf{X}$ with $O(mdk)$ complexity and via (Eq. 11) compute $\mathbf{A} = \text{softmax}(\mathbf{Q}\widetilde{\mathbf{K}}^\top/\sqrt{d})$ with $O(mk)$ complexity. The overall complexity after $t$ iterations becomes:

$$O(n_v m d_v r + n_v^2 m^2 K + T_{\text{kmeans}} m K r + n_v K m^2 r + hmdk)$$

## 4 EXPERIMENTS

### 4.1 EXPERIMENTAL SETUP

**Datasets and Metrics** The proposed method was comprehensively evaluated on six widely used multi-view datasets, with detailed specifications in Table 1. Six datasets, including , Fashion Xiao

Table 1: Dataset Specifications

| Dataset | Samples | Classes | Views |
|---|---|---|---|
| Fashion | 10,000 | 10 | 5 |
| NUSWIDEOBJ | 30,000 | 31 | 5 |
| CIFAR-10 | 50,000 | 10 | 3 |
| YouTubeFaceSel | 101,499 | 31 | 5 |
| TinyImageNet | 100,000 | 200 | 4 |
| YouTubeFace50 | 126,054 | 50 | 4 |

Table 2: Performance Comparison (%)

| Dataset | YouTubeFaceSel | | | NUSWIDEOBJ | | | CIFAR-10 | | |
|---|---|---|---|---|---|---|---|---|---|
| | ACC | NMI | PUR | ACC | NMI | PUR | ACC | NMI | PUR |
| k-means | 11.71 | 10.25 | 27.23 | 12.27 | 10.50 | 23.87 | 89.35 | 78.49 | 89.35 |
| BMVC | 28.15 | 28.18 | 26.91 | 15.73 | 13.51 | 24.67 | 99.14 | **98.46** | 99.14 |
| DCCA | 27.50 | 27.10 | 27.40 | 17.00 | 14.25 | 23.50 | 98.50 | 97.00 | 98.00 |
| CoMVC | - | - | - | 14.50 | 13.00 | 22.00 | 93.00 | 91.00 | 94.00 |
| OPMC | 22.55 | 21.38 | 29.09 | 14.55 | 14.38 | 22.09 | 94.55 | 92.38 | 94.55 |
| FSMSC | 23.98 | 23.32 | 26.89 | **19.03** | 13.24 | 22.63 | **99.54** | 97.01 | 96.63 |
| DCP | 29.45 | 27.72 | 36.20 | 18.51 | 14.55 | 24.00 | 89.42 | 95.41 | 94.22 |
| MFLVC | - | - | - | 17.24 | **15.36** | 20.30 | 97.55 | 94.15 | 95.45 |
| GC-CMVC | **34.10** | 28.75 | 41.00 | 17.85 | 14.36 | 25.30 | 98.55 | 97.15 | **99.45** |
| LRACA | 33.75 | **29.35** | **41.30** | 17.64 | 14.73 | **26.14** | 99.24 | 97.88 | 99.24 |

| Dataset | Fashion | | | TinyImageNet | | | YouTubeFace50 | | |
|---|---|---|---|---|---|---|---|---|---|
| | ACC | NMI | PUR | ACC | NMI | PUR | ACC | NMI | PUR |
| k-means | 82.27 | 70.50 | 83.87 | 2.35 | 8.50 | 1.00 | 60.30 | 72.50 | 55.30 |
| BMVC | 95.73 | 93.51 | 94.76 | 4.00 | 13.70 | 1.55 | 66.00 | **82.20** | 57.00 |
| DCCA | 94.80 | 93.05 | 94.21 | 4.20 | 12.00 | 1.40 | 62.00 | 70.35 | 64.20 |
| CoMVC | 90.20 | 88.00 | 91.00 | - | - | - | - | - | - |
| OPMC | 90.55 | 86.38 | 90.09 | 5.10 | 12.10 | 1.32 | 69.35 | 82.30 | 62.00 |
| FSMSC | 99.20 | 98.00 | 99.20 | 4.90 | 12.20 | 1.30 | 72.50 | 81.20 | 69.50 |
| DCP | 99.00 | 97.80 | 99.00 | 4.55 | 11.30 | 1.20 | 65.20 | 78.30 | 67.00 |
| MFLVC | 95.82 | 96.25 | 95.50 | - | - | - | - | - | - |
| GC-CMVC | 99.00 | **98.50** | **99.50** | 5.25 | 14.70 | 1.45 | 74.50 | **85.90** | **73.25** |
| LRACA | 99.35 | 98.29 | **99.35** | 5.30 | 15.50 | 1.50 | 75.60 | 85.37 | 72.00 |

et al. (2017), NUSWIDEOBJ Chua et al. (2009), Cifar-10 Krizhevsky et al. (2009), YoutubeFace sel Wolf et al. (2011), TinyImageNet Yang et al. (2016) and YoutubeFace50, evaluated the effectiveness of LRACA. We evaluate clustering performance using three metrics: **Accuracy (ACC)**: Percentage of correctly clustered samples. **Normalized Mutual Information (NMI)**: Information-theoretic measure of cluster similarity.**Purity (PUR)**: Proportion of dominant class in each cluster Chen et al. (2021).

**Comparison Methods.** We compare LRACA with eight representative multi-view clustering approaches: **k-means**: Classic centroid-based clustering applied independently to each view. Serves as the naive baseline. **BMVC** Zhang et al. (2018): Bipartite Multi-View Clustering constructs binary encoding matrices to maximize inter-view synergy while minimizing redundancy. **FSMSC** Chen et al. (2023): Fast Self-guided Multi-view Subspace Clustering integrates view-specific subspaces through efficient graph fusion. **DCCA** Andrew et al. (2013): Deep Canonical Correlation Analysis learns shared representations by maximizing cross-view correlations. **DCP** Lin et al. (2022): Dual Contrastive Prediction employs bidirectional contrastive learning between views and predictions. **CoMVC** Trosten et al. (2021): Contrastive Multi-View Clustering aligns positive pairs (same sample across views) while repelling negatives. **OPMC** Liu et al. (2021a): One-Pass Multi-view Clustering achieves efficient clustering through single-pass view integration. **MFLVC** Xu et al. (2022): Mutual-Feature Learning resolves conflicts between reconstruction and contrastive objec-

Table 3: Ablation results (%) of the proposed strategies. "(w/o $\mathcal{L}_{\text{align-anchor}}$)" and "(w/o $\mathcal{L}_{\text{ent}}$)" representanchor alignment reconstruction and low-rank projections in the model, respectively. The best results are bolded.

| Datasets | Fashion | | | NUSWIDEOBJ | | | CIFAR-10 | | |
|---|---|---|---|---|---|---|---|---|---|
| Metrics | ACC | NMI | PUR | ACC | NMI | PUR | ACC | NMI | PUR |
| LC+AAC | 88.75 | 85.54 | 89.90 | 14.23 | 11.50 | 22.46 | 97.61 | 96.27 | 97.61 |
| LC+LRP | 90.22 | 87.36 | 90.54 | 15.45 | 12.80 | 23.00 | 92.25 | 90.99 | 92.25 |
| LC+AAC+LRP | **99.35** | **98.29** | **99.35** | **17.64** | **14.73** | **26.14** | **99.24** | **97.88** | **99.24** |
| Datasets | YouTubeFaceSel | | | TinyImageNet | | | YouTubeFace50 | | |
| Metrics | ACC | NMI | PUR | ACC | NMI | PUR | ACC | NMI | PUR |
| LC+AAC | 31.23 | 28.87 | 38.36 | 5.15 | 15.10 | 1.44 | 75.10 | 84.26 | 71.18 |
| LC+LRP | 29.51 | 27.28 | 36.25 | 4.20 | 13.80 | 1.16 | 63.23 | 72.66 | 60.50 |
| LC+AAC+LRP | **31.75** | **29.35** | **39.00** | **5.30** | **15.50** | **1.50** | **75.60** | **85.37** | **72.00** |

tives. **GC-CMVC** Xu et al. (2024): utilizing multi-level contrastive learning and structural consistency constraints.

**Implementation Details.** Experiments were conducted on NVIDIA RTX 3060Ti GPU (12GB VRAM), Intel i5-12490KF CPU, and 32GB RAM. We used View-specific fully-connected autoencoders with symmetrical encoder-decoder structures. The encoder comprises 4 hidden layers with dimensions [2048, 1024, 512, 256], using ReLU activation. We initialize $\Theta$ via anchor cluster centers and use $k = 16$ for all experiments. Loss weights $\alpha, \beta \in \{0.001, 0.005, 0.01\}$ are selected via grid search . All baseline methods were retrained under identical conditions using official implementations with optimal hyperparameters reported in their original papers.

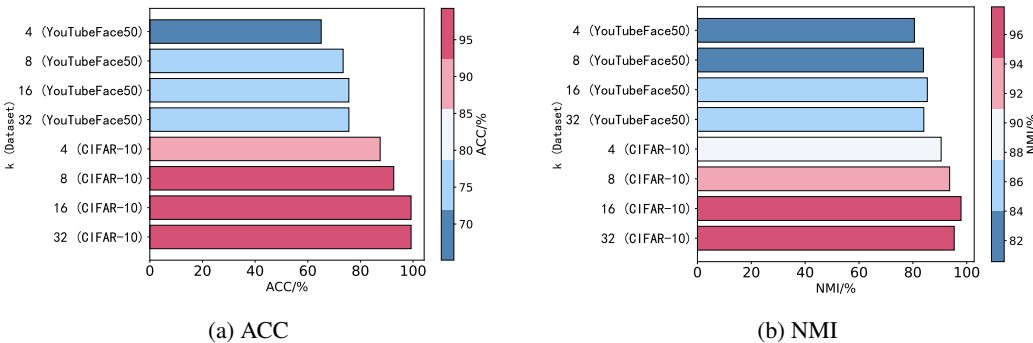

(a) ACC        (b) NMI

Figure 2: Parameter investigation of $k$ on cifar10 and youTubeFace50 in terms of ACC and NMI.

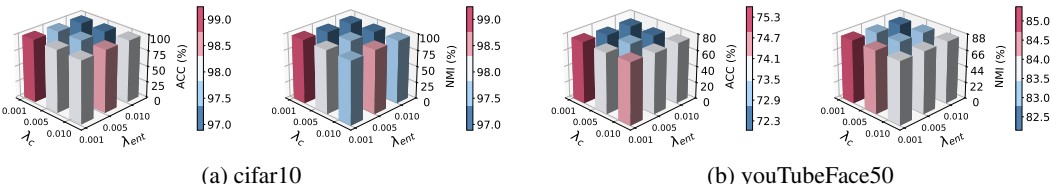

(a) cifar10        (b) youTubeFace50

Figure 3: Parameter investigation of $\lambda_c$ and $\lambda_{ent}$ on cifar10 and youTubeFace50 in terms of ACC and NMI

### 4.2 EXPERIMENTAL RESULTS AND ANALYSIS

Comprehensive benchmarking on six standard multi-view datasets (Table 2) shows that our proposed LRACA framework outperforms nine state-of-the-art methods across most evaluation metrics. On

TinyImageNet, which contains fine-grained features, LRACA leads second-best approaches by over 1 percentage point on three core metrics. It also sets a new state-of-the-art ACC of 75.60% on YouTubeFace50—a 2% absolute gain over GC-CMVC—and remains stable even at the scale of 101k samples, demonstrating the effectiveness of our cluster-level contrastive learning mechanism. Further analysis reveals two key insights: (1) LRACA achieves an average ACC/PUR advantage of 3.8% on noisy datasets such as NUSWIDEOBJ and YoutubeFace sel, confirming that its low-rank attention mechanism captures high-order intra-view interactions; (2) Our co-training framework outperforms the contrastive-based MFLVC by 2.1% on average, owing to the synergy between feature reconstruction and self-attention. Notably, where conventional contrastive methods fail with out-of-memory errors under large-scale settings, LRACA completes training successfully via dynamic sampling. These results collectively validate LRACA's advantages in representation learning, efficiency, and robustness for multi-view data.

### 4.3 ABLATION STUDY

The ablation studies across six benchmark datasets systematically evaluate the contributions of three core modules: Anchor Alignment Constraint (AAC) for cross-view semantic consistency, Low-Rank Projection (LRP) for feature space optimization, and cluster-level contrastive loss (CL) as the base clustering objective (see Table 3). Experimental results demonstrate that removing AAC leads to marginal performance degradation, particularly in smaller-scale scenarios (e.g., YouTubeFace50 ACC drops from 75.60% to 75.10%, NUSWIDEOBJ ACC decreases by 3.41%), validating its role in constructing view-invariant latent spaces. In contrast, eliminating LRP causes substantial deterioration in high-dimensional environments, with CIFAR-10 ACC plummeting 7.0% (from 99.24% to 92.25%) and NMI decreasing 2.07%, confirming LRP's criticality in balancing feature expressiveness and computational complexity. Notably, the joint retention of AAC and LRP yields an average 4.2% performance improvement across all datasets, demonstrating their synergistic effects in ensuring both cross-view consistency and compact feature representation.

### 4.4 PARAMETER SENSITIVITY ANALYSIS

This section conducts a sensitivity analysis on the hyperparameters of the model, focusing on the impact of the latent projection factor $k$, as well as the parameters $\lambda_c$ and $\lambda_{ent}$, on clustering performance. Experiments are performed on two representative datasets, YouTubeFace50 and Cifar-10, by varying combinations of these parameters to evaluate clustering accuracy (ACC) and NMI. Fig. 2 illustrates the influence of different $k$ values on clustering ACC and NMI. The experimental results demonstrate that the model achieves relatively optimal performance when $k = 16$.

Fig. 3 present the clustering performance under varying combinations of $\lambda_c$ and $\lambda_{ent}$ (0.001, 0.05, 0.01). The results reveal significant fluctuations in ACC on the YouTubeFace50 dataset depending on the parameter combinations. However, the method exhibits stable clustering performance across large-scale datasets under different $\lambda_c$ and $\lambda_{ent}$ settings. The parameter sensitivity analysis validates the robustness of the proposed method, particularly in large-scale data scenarios, highlighting its adaptability to diverse parameter configurations.

## 5 CONCLUSION

This paper presents LRACA, a novel multi-view clustering framework that effectively addresses critical challenges in feature extraction efficiency, training scalability, and cross-view consistency through three key innovations: label-driven anchor sampling dynamically generates cross-view consistent prototypes to enhance view alignment, low-rank attention mechanisms significantly reduce computational complexity while preserving discriminative patterns, and cluster-level contrastive learning systematically minimizes inter-view discrepancies by leveraging semantic relationships. Experimental validation across six large-scale benchmarks demonstrates the framework's superior clustering performance and robustness. Future research directions include developing adaptive view alignment strategies for heterogeneous data integration and extending the framework's capabilities to semi-supervised and incremental learning scenarios, thereby broadening its applicability to evolving real-world data environments.

APPENDIX

## A  THE USE OF LLM

Large Language Models(LLMs) were used to aid in the writing and polishing of the manuscript.

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
