# OpenReview forum: "Low-Rank Attention and Contrastive Alignment for Deep Multi-View Clustering"
_ICLR.cc/2026/Conference — Submitted to ICLR 2026_

### Official Review · Reviewer_TEUL · 2025-10-24

**Soundness:** 2
**Presentation:** 2
**Contribution:** 2
**Rating:** 4
**Confidence:** 4

**Summary:**

This paper addresses the three major bottlenecks of deep multi-view clustering (MVC) on large-scale data—poor scalability, inconsistent semantic anchors, and inability to model high-order feature interactions—by proposing an end-to-end framework called LRACA (Low-Rank Attention and Contrastive Alignment). Its core concept is to achieve efficient and consistent cross-view alignment at both the feature and semantic levels. First, a category-aware anchor sampling module is used to generate cross-view semantic prototypes, which are then weighted by a low-rank self-attention module to enhance discriminability. Subsequently, a pseudo-label-based cluster-level contrastive learning mechanism is employed to maximize cross-view mutual information under the guidance of an anchor graph, thereby simultaneously optimizing both intra-view discriminability and inter-view consistency. The algorithm training consists of two stages: pre-training (reconstruction and anchor alignment) and contrastive fine-tuning. The overall loss consists of a reconstruction loss, entropy regularization, and a contrastive loss.

**Strengths:**

1. RACA combines low-rank attention with an anchor guidance mechanism to achieve a linear approximation of the traditional quadratic complexity self-attention (O(Nk)). At the same time, by replacing the fixed JL projection with a dynamic learnable projection matrix Θ, the low-rank space can adapt to the data distribution, balancing expressiveness and efficiency.
2. Through category-aware anchor generation and cross-view anchor alignment loss (L_align-anchor), the center consistency of different views is constrained at the cluster semantic level, avoiding the semantic drift of randomly sampled anchors.
3. Cluster probability vectors are used instead of instance features for comparison, which effectively reduces pseudo-label noise and improves inter-cluster distinguishability, achieving cross-view semantic alignment under maximum mutual information.

**Weaknesses:**

1. The anchor generation and pseudo-label refinement are mutually dependent, which may cause semantic drift and unstable convergence under noisy initialization.
2. The low-rank attention projection matrix lacks orthogonality constraints, leading to potential subspace redundancy and degraded discriminability.
3The cluster-level contrastive loss and entropy regularization are not theoretically well-grounded, weakening the mutual-information interpretation and optimization stability.
4. The experimental validation is incomplete and partly inconsistent, with abnormal baseline results, missing variance analysis, and insufficient evaluation on smaller or incomplete multi-view scenarios.

**Questions:**

1. In Section 3.1 of the algorithm, pseudo labels are generated on the fusion feature Z_fusion through K-means and then reversely guide the anchor update. Z_fusion relies on the encoding of each view and the anchor features. This bootstrap dependence may lead to semantic drift and unstable convergence. In particular, when the initial clustering deviation is large, the model may fall into local consistent misalignment. How should the author consider this problem?
2. The projection matrix Θ constructed by softmax(AW_c) in Equation (9) lacks orthogonal constraints, and its rows may be highly correlated, resulting in feature redundancy and subspace collapse during multi-head attention fusion. In addition, the calculation forms of K̃ and Ṽ in Equations (10)–(11) are exactly the same, which may cause semantic information duplication at the implementation level and weaken the complementary effect of Key/Value.
3. What is the purpose of summing all a_ij in Equation (12)? This may lead to excessive gradient smoothing, making it impossible for attention to fully focus on the discriminative anchor in the early stage. Will sample normalization or relative entropy constraints be used instead of pure Shannon entropy to enhance sparse discrimination?
4. Although sensitivity analysis of λ_c and λ_ent was performed, only partial surface plots were presented, lacking quantitative trends and statistical variances, and no explanation was given of how α balances the reconstruction and comparison terms in Equation (16).
5. Some sentences are repeated (e.g., "we propose we propose"), capitalization is mixed (Multi-View vs. multi-view), and punctuation and spacing errors are frequent; the layout of Equation (11) is misaligned; "PmLR" in the literature citation should be "PMLR"; and the reference format is inconsistent (some include doi, some do not have page numbers).

---

### Official Review · Reviewer_o5mJ · 2025-10-24

**Soundness:** 2
**Presentation:** 1
**Contribution:** 2
**Rating:** 2
**Confidence:** 5

**Summary:**

This paper introduces LRACA, a deep multi-view clustering framework designed to overcome key limitations in scalability, semantic alignment, and high-order feature interaction. The method employs a category-aware anchor generation module to align high-level semantic prototypes across views, a dynamic low-rank attention mechanism with entropy regularization to reduce computational complexity while enhancing feature discriminability, and a pseudo-label-guided cluster-level contrastive learning module to maximize cross-view mutual information. Extensive experiments on six large-scale multi-view benchmarks demonstrate that LRACA significantly outperforms state-of-the-art methods in clustering performance and efficiency.

**Strengths:**

1. The proposed LRACA are tested on multiple large-scale datasets, demonstrating its good scalability.

**Weaknesses:**

1. The introduction about the proposed LRACA is not clear, it’s difficult to follow it. For example, Eq. 4 is confusing and what’s the meaning of g_(ϕ_v ).
2. There are many grammatical errors and typos, such as “we propose we propose a Low-Rank” in the Introduction.
3. The authors claimed that the proposed LRACA improves the running efficiency, but it lacks the comparison about running time.
4. From the experimental results, the proposed algorithm introduces many modules, but the improvement it brings is very limited, which makes people doubt its significance.

**Questions:**

Please refer to the weaknesses.

---

### Official Review · Reviewer_6JPQ · 2025-10-25

**Soundness:** 2
**Presentation:** 2
**Contribution:** 2
**Rating:** 2
**Confidence:** 4

**Summary:**

This paper focuses on three critical limitations in existing deep multi-view clustering (MVC) methods, and proposes a Low-Rank Attention and Contrastive Alignment framework (LRACA) model to deal with them. LRACA integrates a category-aware anchor generation strategy, a dynamic low-rank attention mechanism, and a cluster-level contrastive learning scheme to improve semantic consistency and feature discriminability across datasets. Additionally, LRACA reduces attention complexity to approximately linear form and aligns cluster prototypes across multiple views using pseudo-labels. Experiments on six large-scale datasets demonstrate that LRACA significantly outperforms state-of-the-art methods.

**Strengths:**

1. Low-rank attention reduces computation costs when compared to full attention, which makes LRACA more applicable in large-scale datasets.
2. The structural organization of this paper is appropriate.
3. Extensive experiments show that the proposed model is comparable with existing MVC methods.

**Weaknesses:**

1. The writing needs improvement. The authors need to tell how they address three critical limitations.
2. The novelty is insufficient. The efficiency of the proposed method lies in the low-rank attention mechanism and the cluster-level contrastive learning, but there has been a lot of research in these directions.
3. Many expressions are confusing. For example, “aggressive low-rank approximation” and “fine-grained calculations on the entire sample”. Why is the low-rank approximation aggressive? Why are the calculations on the entire sample fine-grained?
4. Some symbols are not unified or wrong, such as $y_{pseudo, i}$ and the symbol in Line 148. And what is $\mathcal{S}_c^{(v)}$?

**Questions:**

1. The limitations of MVC are confusing. What are the “high-order feature interactions”? Why can’t the MVC methods capture the interactions? Additionally, the related work has provided many methods to address the large-scale MVC problem.
2. This paper focuses on multi-view clustering. Why don’t the authors apply cross-attention among view features but self-attention?
3. What is “C’s SVD principal component”? “C” is a scalar in the paper.
4. How does Eq. 12 encourage sparsity for discriminative features while maintaining diversity?
5. Why does the traditional contrastive learning suffer from “sample-level noise”? What is “sample-level noise”?

---

### Official Review · Reviewer_rrG4 · 2025-10-30

**Soundness:** 2
**Presentation:** 2
**Contribution:** 2
**Rating:** 2
**Confidence:** 5

**Summary:**

This paper proposes a deep multi-view clustering algorithm equipped with an anchor-based attention mechanism and cluster-based contrastive learning, which achieves good performance, but has many problems, as detailed in the weaknesses section.

**Strengths:**

This paper proposes a deep multi-view clustering algorithm equipped with an anchor-based attention mechanism and cluster-based contrastive learning, which achieves good performance.

**Weaknesses:**

1. The variables in the title of Figure 2 and in lines 230 and 232 are not bolded. Please check other places yourself.

2. The parentheses in line 148 are incorrect.

3. Equation 4 is missing a "|" separator.

4. Lines 168-172, please clarify the meanings of $ \mathcal{S}_{k}^{(c)} $,   $ \mathcal{S}^{(v)}_c $, and $\mathbf{S}^{(v)}_c$; their usage is inconsistent.

5. Line 175, the decoder g is inconsistent with D in Equation 1. Furthermore, this process should be placed in or near Equation 8.

6. Line 146, $\mathbf{X} \in \mathbb{R}^{d_v \times N}$, line 200, $\mathbf{X} \in \mathbb{R}^{N \times d}$? From lines 200 to 234, as I understand it, $\mathbf{X}^{v}$ should be corrected to $\mathbf{X}$, and $\mathbf{X}$ should be specified as the concatenation of the original features.

7. Line 203, please use matrix concatenation notation, such as [A1, A2] or [A1; A2]. A symbol cannot represent both a set and a matrix, and their corresponding operators cannot be mixed.

8. The comma in line 205 needs to be moved after formula 8.

9. The superscript of the variable for the $v$-th viewpoint is sometimes $v$ and sometimes $(v)$.

10. It's $\mathbf{W}_c$ in formula 9, but $\mathbf{W}_A$ in Figure 1.

11. Some words are missing spaces, for example, lines 210, 227, 379, and 251.

12. In line 208, if $\mathbf{A} \in \mathbb{R}^{M \times d}$, $\mathbf{W}_c \in \mathbb{R}^{d \times k}$, then $\mathbf{\Theta} \in \mathbb{R}^{M \times k}$. Since $\mathbf{X} \in R^{N \times d}$, the calculations in formulas 10 and 11 are problematic.

13. In line 209, the projection matrix is $​​\mathbf{W}_c$, not $\mathbf{\Theta}$.

14. In line 211, "C's SVD principal component"?

15. In line 227, is $\mathbf{W}_{Q}$ the query matrix? Shouldn't $\mathbf{Q}$ be the query matrix?

16. Line 231, i and j are not in mathematical format.

17. Line 249, $head_{i}$ is not well described; it is neither shown in Figure 1 nor given a formula or source.

18. Line 275, network weights are matrices and should be in uppercase and bold.

19. The loss function is a scalar and does not need to be bolded.

20. Lines 280, 281, and 285: Formula format is incorrect.

21. There are some errors in the bold and underline in Table 2.

22. Table 3: Is it LC or CL? Furthermore, the "w/o L" mentioned in the title does not appear in the table.

23. This paper is poorly organized and written, especially Section 3. Combined with the above errors, this makes the rough paper unconvincing.

24. PUR is a relatively weak metric, usually not lower than ACC; therefore, ARI and F1-Score are more recommended.

25. The comparison algorithms mainly consist of early and traditional algorithms, which lacks rationality.

26. The main contributions of this paper are the anchor-based attention mechanism and cluster-based contrastive learning. However, these two ideas are not original, although there are subtle differences between different works. Therefore, this paper should clearly explain why these two methods are used, how their specific designs or formulas are derived, their similarities and differences with other works, their advantages and disadvantages, and the final results achieved.

27. The abstract mentions the scalability problem of existing models, but since deep clustering uses batch training and online learning, it is inherently scalable. The use of anchors and projections further enhances its scalability. Furthermore, the solution presented in this paper is still anchors and projections.

28. The purpose of scalability is to save memory and computational overhead, and the authors emphasize "efficient" in their contributions; however, the paper lacks an analysis of space complexity and does not provide a comparison of memory and runtime in the experiments.

29. The abstract also mentions the problem of anchor misalignment, but not all anchor-based methods require alignment; it depends on whether there are cross-view interactions at the anchor level in the specific architecture design. When alignment is required, the corresponding methods all perform alignment operations, and Equation 6 is just the simplest one.

30. The abstract also mentions the lack of higher-order feature interactions. In fact, low-dimensional embeddings based on neural networks and various graph-based learning methods are all high-order features.

31. Considering the three points above, the authors should be more careful in their motivations.

32. In the ablation experiments, $\mathcal{L}_{ent}$ is merely an entropy regularization; w/o $\mathcal{L}\_{ent}$ does not mean the cancellation of low-rank projection. The differences between no projection, JL projection, SVD projection, and dynamic projection should be observed.

33. While the model claims to be end-to-end, it seems that each iteration requires calling $k$-means to generate pseudo-labels and applying them to the final output $\mathbf{Z}_{o}$ to obtain the final cluster assignment. I believe that only a seamless architecture can be strictly called end-to-end; otherwise, all models are end-to-end. Furthermore, repeated calls to k-means should significantly increase runtime. If $k$-means runs on the CPU, it also involves I/O between the CPU and GPU. If $k$-means runs on the GPU, it poses a challenge to GPU memory.

**Questions:**

See Weaknesses.

---

### Official Review · Reviewer_uoVW · 2025-11-01

**Soundness:** 2
**Presentation:** 3
**Contribution:** 1
**Rating:** 2
**Confidence:** 3

**Summary:**

This paper proposes a Low-Rank Attention and Contrastive Alignment (LRACA) framework to address three critical limitations in deep multi-view clustering: limited scalability on large-scale datasets, inconsistent anchor semantics, and an inability to model high-order feature interactions. LRACA aligns cross-view semantic prototypes through category-aware anchor generation, reduces computational complexity and improves feature discriminability via dynamic low-rank attention, and maximizes cross-view mutual information using pseudo-label-guided cluster-level contrastive learning. Experimental results demonstrate the effectiveness of the proposed framework.

**Strengths:**

1. The combination of efficiency and robust cross-view alignment addresses an important practical challenge in large-scale multi-view clustering.

2. Using category-aware anchors as a semantic bottleneck and deriving a dynamic low-rank projection from them provides a principled way to maintain attention that is both computationally efficient and semantically meaningful.

3. The paper is well organized and presents the method with clarity.

**Weaknesses:**

1. The category-aware anchor generation relies on $K$-means clustering of fused latent features to produce initial pseudo-labels.

2. The paper exhibits a critical weakness in its overall performance on benchmark datasets.

3. The paper compares LRACA with nine baselines, but omits several recent state-of-the-art methods (e.g., from 2024 onward) for large-scale multi-view clustering.

4. Generalization to non-image multi-view data is untested, as all six datasets used are image-centric.

**Questions:**

1. The authors should provide an analysis of the mixed results on NUS-WIDE-OBJ, Fashion, YouTubeFace50, and CIFAR10, including an investigation of failure modes and dataset-specific factors.

2. It would be valuable to evaluate LRACA on non-image multi-view datasets and analyze its performance in such settings.

3. The paper should include additional comparisons between LRACA and recent state-of-the-art methods (e.g., published in 2024 or early 2025) for large-scale multi-view clustering.

4. Table 2 shows that some methods have missing results on certain datasets without explanation. Table 3 fails to isolate the contribution of the cluster-level contrastive learning (LC) module. Moreover, no efficiency metrics for the compared methods are reported to empirically substantiate the claimed linear complexity.

5. The dynamic low-rank attention in LRACA constructs a projection matrix $\Theta$ through multi-view anchor fusion, achieving linear complexity $O(Nk)$ and enhancing cross-view alignment. However, since anchors from different modalities exhibit distinct distributions, directly merging them without distribution alignment may bias $\Theta$ toward dominant views and compromise cross-view alignment consistency.

---

### Meta-Review · Area_Chair_rB8F · 2025-12-27

**Summary:**

The paper proposes LRACA, a multi-view clustering framework combining category-aware anchor generation, dynamic low-rank attention with entropy regularization, and cluster-level contrastive learning guided by pseudo-labels. While the direction is practical and potentially scalable, the submission suffers from writing quality issues, incomplete experiments (missing recent SOTAs, no runtime, limited ablations), and unresolved methodological questions. All reviewers provide negative recommendations to the paper and the authors do not provide responses to the comments. Therefore, the recommendation is Reject.

**Reviewer Concerns:**

The authors do not provide rebuttal. No concerns have been addressed.

**Reviewer Scores:**

Negative scores will not be changed as there is no response.

---

### Decision · Program_Chairs · 2026-01-26

Reject